# EXTENDING DIFFERENTIAL TEMPORAL DIFFERENCE METHODS FOR EPISODIC PROBLEMS

## ABSTRACT

Differential temporal difference (TD) methods are value-based reinforcement learning algorithms that have been proposed for infinite-horizon problems. They rely on reward centering, where each reward is centered by the average reward. This keeps the return bounded and removes a value function's state-independent offset. However, reward centering can alter the optimal policy in episodic problems, limiting its applicability. Motivated by recent works that emphasize the role of normalization in streaming deep reinforcement learning, we study reward centering in episodic problems and propose a generalization of differential TD. We prove that this generalization maintains the ordering of policies in the presence of termination, and thus extends differential TD to episodic problems. We show equivalence with a form of linear TD, thereby inheriting theoretical guarantees that have been shown for those algorithms. We then extend several streaming reinforcement learning algorithms to their differential counterparts. Across a range of base algorithms and environments, we empirically validate that reward centering can improve sample efficiency in episodic problems.

## 1 INTRODUCTION

The average reward formulation of reinforcement learning (Mahadevan, 1996)—which can be described as an undiscounted objective for continuing problems—has led to the development of algorithms that shift rewards by the average reward (Schwartz, 1993; Sutton & Barto, 2018; Wan et al., 2021). This mean-centering of rewards prevents the undiscounted, infinite sum of rewards from diverging. Temporal difference (TD) methods which predict this sum of centered rewards form the *differential* TD family of algorithms. Recent work separated centering from the average reward formulation by demonstrating its utility in discounted problems (Naik et al., 2024; Naik, 2024)—a setting where centering is not necessary for bounding an infinite sum of rewards. However, its use remains limited to continuing problems because in episodic problems, the ordering of policies is not preserved when rewards are shifted. To illustrate this, consider an episodic problem where some positive constant $c$ is subtracted from every reward. Subtracting a sufficiently large $c$ produces optimal behavior which terminates as quickly as possible. Conversely, adding a sufficiently large $c$ to every reward encourages behavior that prolongs the episode (i.e., avoids termination).

Normalization methods have recently garnered interest in deep reinforcement learning (e.g., Lyle et al., 2023; Lyle et al., 2024; Palenicek et al., 2025). Notably, normalization has shown substantial benefit in *streaming* deep reinforcement learning (Vassan et al., 2024; Elsayed et al., 2024)—the buffer-free, online, incremental learning setup of the original reinforcement learning algorithms (Sutton, 1988b; Sutton & Barto, 2018). Much of the recent work on normalization has focused on techniques such as input centering and scaling (Sutton, 1988a), layer normalization (Ba et al., 2016), and output scaling. However, less attention has been paid to output centering, with the lack of optimal policy invariance in episodic problems cited as the concern with it (Lee et al., 2025). Episodic environments are widely used for evaluation (e.g., Young & Tian, 2019; Towers et al., 2024), motivating a revisit of differential TD and exploring whether its applicability can be expanded.

In this work, we introduce a strict generalization of differential TD that extends its applicability to both discounted and undiscounted episodic problems. Through the lens of potential-based reward shaping, we prove that the modification maintains invariance of the optimal policies. We further show an equivalence between differential TD and a state-and-action-independent, output-level bias

unit, establishing that the algorithm shares theoretical guarantees (e.g., convergence to the fixed point) with those of TD with linear function approximation. In tabular episodic problems, we highlight the utility of centering and identify scenarios where we might expect improvement. Finally, in the streaming deep reinforcement learning setting, we show that our generalization of differential TD integrates seamlessly into existing algorithms, scales effectively to non-linear function approximation, and preserves the sample complexity benefits previously observed in continuing problems.

## 2 BACKGROUND AND RELATED WORK

Reinforcement learning is typically formalized as a Markov decision process (MDP), characterized by a set of states $\mathcal{S}$, sets of each state's available actions $\mathcal{A}(s)$, and an environment transition model $p(s', r|s, a) = P(S_{t+1} = s', R_{t+1} = r|S_t = s, A_t = a)$. For each discrete time step $t$, an agent observes its current state $S_t \in \mathcal{S}$, selects an action $A_t \in \mathcal{A}(S_t)$, and jointly samples a next state $S_{t+1} \in \mathcal{S}$ and reward $R_{t+1} \in \mathbb{R}$ according to the environment transition model. Actions are selected according to a policy $\pi(a|s) = P(A_t = a|S_t = s)$, and reinforcement learning agents in control problems aim to find the optimal policy $\pi^*$ which maximizes a reward-based objective. A common objective is to maximize the expected discounted return. The return is given by:

$$G_t \stackrel{\text{def}}{=} \sum_{k=0}^{T-t-1} \gamma^k R_{t+k+1},$$

where $\gamma \in [0, 1]$ and $T$ being an episode's final time-step, or $\gamma \in [0, 1)$ and $T = \infty$ in infinite-horizon, continuing problems. Value-based methods for reinforcement learning compute or approximate *value-functions*, which are defined to be expected returns conditioned on a state (or state-action pair) under a policy $\pi$:

$$v_\pi(s) \stackrel{\text{def}}{=} \mathbb{E}_\pi[G_t|S_t = s], \forall s$$

$$q_\pi(s, a) \stackrel{\text{def}}{=} \mathbb{E}_\pi[G_t|S_t = s, A_t = a], \forall s, a,$$

with $v_\pi(s)$ denoted the *state-value* function and $q_\pi(s, a)$ denoted the *action-value* function. The process of computing a policy's value function is referred to as *policy evaluation*. Such values may then inform decisions via *policy improvement*—a theorem stating that behaving greedily with respect to $q_\pi$ will result in an improved policy $\pi'$ where $q_{\pi'}(s, a) \geq q_\pi(s, a), \forall s, a$. Policy evaluation and improvement can then alternate in a process of *policy iteration* to approach an optimal policy.

A popular approach to policy evaluation makes use of a value-function's Bellman equation, where a decision point's value is expressed in terms of successor decision point values. For example, for $v_\pi$:

$$v_\pi(s) = \sum_a \pi(a|s) \sum_{s',r} p(s', r|s, a)\big(r + \gamma v_\pi(s')\big), \forall s.$$

Given a transition $(S_t, A_t, R_{t+1}, S_{t+1})$, temporal difference (TD) methods (Sutton, 1988b) form a sample-based estimate of $v_\pi(S_t)$ based on its Bellman equation and take a step toward this target:

$$V(S_t) \leftarrow V(S_t) + \alpha\big(R_{t+1} + \gamma V(S_{t+1}) - V(S_t)\big),$$

where $V \approx v_\pi$ is a learned, approximate value function and $\alpha \in [0, 1]$ is the step-size.

An alternative to the discounted objective is the average reward criterion (Mahadevan, 1996), where an agent seeks to maximize its reward per step from some starting state $S_0$:

$$r(\pi, s) \stackrel{\text{def}}{=} \lim_{n \to \infty} \frac{1}{n} \sum_{t=1}^{n} \mathbb{E}[R_t|S_0 = s, A_{0:t-1} \sim \pi], \forall s, \pi.$$

A unichain assumption is typically made on the MDP, making $r(\pi, s)$ independent of state and simplifying our notation to $r(\pi)$. This objective is akin to maximizing an undiscounted return in an infinite-horizon, continuing setting. Standard value-based methods are not applicable here as undiscounted, infinite-horizon returns are generally infinite. Value-based, average reward algorithms instead work with *differential* returns:

$$G_t^\Delta \stackrel{\text{def}}{=} \sum_{k=0}^{\infty} \big(R_{t+k+1} - r(\pi)\big),$$

where the average reward is subtracted from each reward to ensure the sum converges. Given corresponding differential value functions (e.g., $v_\pi^\Delta(s) \overset{\text{def}}{=} \mathbb{E}_\pi[G_t^\Delta | S_t = s]$), a TD method for this setting maintains an estimate of its average reward (which we denote $b$) and uses this to estimate the differential return:

$$b \leftarrow b + \eta\alpha\big(R_{t+1} - b\big)$$

$$V^\Delta(S_t) \leftarrow V^\Delta(S_t) + \alpha\big(R_{t+1} - b + V^\Delta(S_{t+1}) - V^\Delta(S_t)\big),$$

where $\eta \in [0, 1]$ produces an effective step size of $\eta\alpha$ for the update to $b$, which usually has a slower time-scale. This algorithm which directly averages sampled rewards is called R-learning (Schwartz, 1993). This was later improved upon by Wan et al. (2021) with the differential TD algorithm:

$$\delta = R_{t+1} - b + V^\Delta(S_{t+1}) - V^\Delta(S_t) \tag{1}$$

$$b \leftarrow b + \eta\alpha\delta$$

$$V^\Delta(S_t) \leftarrow V^\Delta(S_t) + \alpha\delta$$

In addition to better empirical performance, updating $b$ using the value update's error allows for *off-policy* estimation of the average reward. That is, $b$ converges to the average reward of the policy being evaluated, allowing it to differ from that which chooses actions.

Recent work by Naik et al. (2024) reintroduced $\gamma$ into Equation 1, decoupling differential TD's reward centering mechanism from the average reward objective and demonstrating its utility on discounted objectives. This extension was motivated by removing an often large, state-independent offset in the value function that is evident in a value function's Laurent series decomposition:

$$v_\pi(s) = \frac{r(\pi)}{1 - \gamma} + v_\pi^\Delta(s) + e_\pi(s, \gamma), \forall s,$$

where $v_\pi(s)$ is a discounted value function, $v_\pi^\Delta(s)$ is an undiscounted differential value function, and $e_\pi(s, \gamma)$ is an error term that captures the difference between discounted and undiscounted values (and vanishes as $\gamma \to 1$). Subtracting $r(\pi)$ from each reward in a discounted, infinite-horizon return results in a subtraction of $\frac{r(\pi)}{1-\gamma}$ from the return, thus canceling the constant in the above decomposition. Reward centering was shown to improve sample efficiency but remained limited to continuing problems. In episodic problems, the shift in return from shifts in reward depends on the remaining episode length. Because the remaining episode length varies across states and actions, invariance of the optimal policies is not guaranteed.

Interestingly, differential TD is a possible explanation for the interplay between phasic and tonic dopamine in the brain (Gershman et al., 2024). This biological plausibility further motivates developing and understanding centered TD algorithms.

## 3 CENTERING REWARDS IN THE PRESENCE OF TERMINATION

In this section, we demonstrate how to maintain invariance of the optimal policies under reward centering. In particular, we consider a view of reward centering as potential-based reward shaping (Ng et al., 1999). Given some function $F(s, a, s')$ of the form:

$$F(s, a, s') = \gamma\Phi(s') - \Phi(s),$$

where $\Phi(S_T) \overset{\text{def}}{=} 0$, adding $F(s, a, s')$ to each reward maintains invariance of the optimal policies while having an effect on learning dynamics. Without assumptions on the MDP, $r(s, a, s') + F(s, a, s')$ was shown to be the only reward transformation with this property (Ng et al., 1999). For some free variable $b$, if we define $\Phi(s)$:

$$\Phi(s) \overset{\text{def}}{=} \frac{b}{1 - \gamma},$$

we get the following state-independent reward shaping term:

$$F(s, a, s') = \gamma\frac{b}{1 - \gamma} - \frac{b}{1 - \gamma}$$

$$= b\frac{\gamma - 1}{1 - \gamma}$$

$$= -b.$$

This produces a constant shift in reward. If $b$ estimates the average reward, this shaping term in continuing problems recovers differential TD and validates that the ordering of policies remains unchanged. However, reward shaping makes no assumption about the problem setting—if we recognize that $\Phi(s)$ must be zero at terminal states (Grześ, 2017), we get:

$$F(s, a, s') = \begin{cases} \frac{-b}{1-\gamma}, & \text{if } s' \text{ is terminal} \\ -b, & \text{otherwise} \end{cases}$$

This leads to the following TD updates:

$$V^{\Delta}(S_t) \leftarrow \begin{cases} V^{\Delta}(S_t) + \alpha\big(R_{t+1} - \frac{b}{1-\gamma} - V^{\Delta}(S_t)\big), & \text{if } s' \text{ is terminal} \\ V^{\Delta}(S_t) + \alpha\big(R_{t+1} - b + \gamma V^{\Delta}(S_{t+1}) - V^{\Delta}(S_t)\big), & \text{otherwise} \end{cases}$$

which can be equivalently expressed through the following terminal differential value definition:

$$V^{\Delta}(S_t) \leftarrow V^{\Delta}(S_t) + \alpha\big(R_{t+1} - b + \gamma V^{\Delta}(S_{t+1}) - V^{\Delta}(S_t)\big)$$

$$V^{\Delta}(S_T) \stackrel{\text{def}}{=} \frac{-b}{1-\gamma}.$$

The intuition behind this terminal value definition lies in the equivalence between a terminal state and an infinitely self-looping state with zero reward. The update is akin to transforming an episodic problem into an equivalent, hypothetical continuing problem—a setting where constant shifts in reward lead to constant shifts in return. In this view, the infinite discounted shifts in the self-looping state are summarized with a closed-form expression. We note, however, that this episodic-to-continuing transformation is from the *perspective of the value function*, as an agent still resets to a starting state and does not perform updates in the terminal state.

Reward shaping also has an equivalence with value-function initialization (Wiewiora, 2003), suggesting that it can influence exploration via means like optimistic initialization (Sun et al., 2022). The relationship between reward shaping and value-function initialization provides insight as to why we might expect centering to improve sample efficiency. It is akin to initializing a value function to its mean and reducing the distance that each state- or action-value has to travel. It is not an exact equivalence here, as $b$ changes over time (Devlin & Kudenko, 2012). However, because we are estimating a single scalar, it is a relatively simple learning problem.

Because the modification is equivalent to defining a terminal differential value, formally this is a generalization of differential TD as the algorithm previously did not intend to encounter termination. However, because of the division by $1 - \gamma$ in the terminal differential value, the above modification does not apply to undiscounted, episodic problems.

## 4    LEARNING EPISODIC DIFFERENTIAL VALUES

The previous section detailed how optimal policy invariance can be maintained when centering rewards in episodic problems. However, the reward shaping perspective assumes the potential function is fixed and does not suggest if the algorithm is sound if $b$ is continually updated. To reconcile this, we view differential TD as learning values where the value function has an output-level bias unit that is independent of state and action. To establish this equivalence, we define a value function to be the sum of differential values (parameterized by $\mathbf{w}$) and a *bias unit* $b$:

$$V(s; \mathbf{w}, b) \stackrel{\text{def}}{=} V^{\Delta}(s; \mathbf{w}) + b.$$

With a mean-squared-value-error objective and (sample-based) gradient-descent updates (e.g., Sutton & Barto, 2018; Mnih et al., 2015), we get:

$$J(\mathbf{w}, b) \stackrel{\text{def}}{=} \frac{1}{2} \sum_s d(s)\big(v_{\pi}(s) - V(s; \mathbf{w}, b)\big)^2$$

$$\mathbf{w}_{t+1} \leftarrow \mathbf{w}_t + \alpha\big(v_{\pi} - V(S_t; \mathbf{w}_t, b_t)\big)\nabla_{\mathbf{w}}V^{\Delta}(S_t; \mathbf{w}_t)$$

$$b_{t+1} \leftarrow b_t + \eta\alpha\big(v_{\pi} - V(S_t; \mathbf{w}_t, b_t)\big),$$

where—to emphasize the relationship with differential TD—we again specify $\eta \in [0, 1]$ to produce a slower time-scale, effective step-size of $\eta\alpha$ for the bias unit update. Substituting a TD estimate of $v_\pi$ then gives us the following error:

$$
\begin{aligned}
v_\pi - V(S_t; \mathbf{w}_t, b_t) &= R_{t+1} + \gamma V(S_{t+1}; \mathbf{w}_t, b_t) - V(S_t; \mathbf{w}_t, b_t) \\
&= R_{t+1} + \gamma V^\Delta(S_{t+1}; \mathbf{w}_t) + \gamma b_t - V^\Delta(S_t; \mathbf{w}_t) - b_t \\
&= R_{t+1} - (1 - \gamma)b_t + \gamma V^\Delta(S_{t+1}; \mathbf{w}_t) - V^\Delta(S_t; \mathbf{w}_t)
\end{aligned}
$$

This resembles, but does not completely match differential TD (as defined by Equation 1) in that it has an extra $\gamma b_t$ term. However, because the additional term only involves a state- and action-independent scalar, we can use reparameterization to show that this update is equivalent to differential TD if we allow the bias to use a separate step size (as is the case with differential TD). Define $\hat{b} \overset{\text{def}}{=} (1 - \gamma)b$ and $\hat{\eta} \overset{\text{def}}{=} \eta(1 - \gamma)$:

$$
\mathbf{w}_{t+1} \leftarrow \mathbf{w}_t + \alpha\big(R_{t+1} - \hat{b}_t + \gamma V^\Delta(S_{t+1}; \mathbf{w}_t) - V^\Delta(S_t; \mathbf{w}_t)\big)\nabla_{\mathbf{w}} V^\Delta(S_t; \mathbf{w}_t)
$$

$$
b_{t+1} \leftarrow b_t + \eta\alpha\big(R_{t+1} - \hat{b}_t + \gamma V^\Delta(S_{t+1}; \mathbf{w}_t) - V^\Delta(S_t; \mathbf{w}_t)\big)\frac{\partial \hat{b}_t}{\partial b_t}
$$

$$
\Leftrightarrow b_{t+1} \leftarrow b_t + \eta\alpha\big(R_{t+1} - \hat{b}_t + \gamma V^\Delta(S_{t+1}; \mathbf{w}_t) - V^\Delta(S_t; \mathbf{w}_t)\big)(1 - \gamma)
$$

$$
\Leftrightarrow b_{t+1} \leftarrow b_t + \hat{\eta}\alpha\big(R_{t+1} - \hat{b}_t + \gamma V^\Delta(S_{t+1}; \mathbf{w}_t) - V^\Delta(S_t; \mathbf{w}_t)\big)
$$

It is evident that if we set $\eta$ and initialize $b_0$ appropriately, and we perform updates on the same sequence of transitions, the updates to $\mathbf{w}$ exactly match those of differential TD. The bias-unit step-size can also be treated as the bias unit's activation value. This interpretation establishes an equivalence with a specific choice of feature representation, and as a result, the analysis of linear TD with discounting (or eventual termination) extends toward differential TD in episodic problems. See Appendix E for a complete proof of convergence which validates this.

The presence of the additional $\gamma b_t$ term prior to reparameterization results in bootstrapping off of uncentered values (i.e., $V$ and not $V^\Delta$). This allows us to define the sum $V^\Delta(S_T; \mathbf{w}) + b \overset{\text{def}}{=} 0$ (or $V^\Delta(S_T; \mathbf{w}) \overset{\text{def}}{=} -b$) to handle terminal states, which is what we get if we substitute $\hat{b}$ into the terminal differential value definition from Section 3. This highlights that the additional $\gamma b_t$ is what follows from a potential function $\Phi(s) = b$. While the two forms are equivalent through the separate step size, it is notably a form which is applicable in episodic problems with $\gamma = 1$. As an example, Algorithm 1 details how we can extend differential Q-learning to handle episodic problems. It provides two forms of the update which, when $\gamma = 1$, must be selected based on whether the problem is known to be continuing or episodic. Either form is applicable when $\gamma < 1$, and as shown above, are formally equivalent under corresponding parameter settings.

The connection with a choice of feature representation highlights that $b$ and $V^\Delta(s; \mathbf{w})$ are jointly optimized under a common objective. This may contrast intuition from the average reward setting where it is often presented as two interacting processes: an average reward estimate which depends on the policy derived from differential value estimates, and differential value estimates which depend on the average reward estimate. This view also presents an interpretation of $\eta$ as balancing credit assignment, which—on a problem dependent basis—may not need to be on a slower time scale.

The bias unit perspective further suggests what $b$ converges to in episodic problems. It is less informative to consider average reward because any policy which eventually terminates has zero average reward due to the equivalence between terminal states and infinite self-loops with zero reward. Because updates are not performed to the values of terminal states, the differential values are centered over non-terminal states, making $b$ approach the expected state-value over the (non-terminal) visitation distribution, subdivided over the expected remaining episode length: $\mathbb{E}_{s \sim d_\pi}[V(s)\frac{1-\gamma}{1-\gamma^{T(s)}}]$, where $d_\pi$ represents normalized expected state visitation counts under policy $\pi$ and $T(s)$ is the expected remaining episode length from state $s$ (See Appendix C).

---

**Algorithm 1** (Generalized) Differential Q-learning

---

Initialize weights $\mathbf{w} \in \mathbb{R}^d$ arbitrarily
Initialize $b \in \mathbb{R}$ arbitrarily
**for** each episode **do**
    $s \sim p(s_0)$
    **for** each step of episode **do**
        $a \sim \pi(\cdot|s)$
        $s', r \sim p(s', r|s, a)$
        **if** $s'$ is terminal **then**
            $\delta \leftarrow r - \frac{b}{1-\gamma} - Q^\Delta(s, a; \mathbf{w})$                       $\triangleright$ $\gamma < 1$
            $\delta \leftarrow r - b - Q^\Delta(s, a; \mathbf{w})$                         $\triangleright$ $\gamma < 1$ or $\gamma = 1$, episodic
        **else**
            $\delta \leftarrow r - b + \gamma \max_{a'} Q^\Delta(s', a'; \mathbf{w}) - Q^\Delta(s, a; \mathbf{w})$     $\triangleright$ $\gamma < 1$ or $\gamma = 1$, continuing
            $\delta \leftarrow r - (1 - \gamma)b + \gamma \max_{a'} Q^\Delta(s', a'; \mathbf{w}) - Q^\Delta(s, a; \mathbf{w})$     $\triangleright$ $\gamma < 1$ or $\gamma = 1$, episodic
        **end if**
        $\mathbf{w} \leftarrow \alpha \delta \nabla_w Q^\Delta(s, a; \mathbf{w})$
        $b \leftarrow \eta \alpha \delta$
        $s \leftarrow s'$
    **end for**
**end for**

---

## 5 EMPIRICAL EVALUATION

To see whether the benefits of centering in continuing problems (Naik et al., 2024) can be achieved in episodic problems, we consider differential Q-learning (Watkins, 1989) with our novel terminal differential value definition (Algorithm 1) and compare it with vanilla, uncentered Q-learning in a 10×10 episodic grid world where the top-left is the start state and the bottom-right is terminal. The grid world uses 4-directional movement where attempting to leave the grid keeps the agent in place. To gain insight into when centering is useful, we consider two reward distributions: $-1$ per step (the *painful* grid world) and 0 per step with 1 upon termination (the *sparse* grid world). Based on the intuition around centering reducing the total distance that outputs need to travel, our hypothesis is that centering will provide substantially more benefit in the painful grid world, since the values deviate more across states. We fixed $\gamma = 0.9$, tuned $\alpha$ for Q-learning, and we tuned $\alpha$ and $\eta$ for differential Q-learning. An $\epsilon$-greedy policy was used for both algorithms with $\epsilon = 0.1$. Full details of the parameter sweeps can be found in Appendix A.

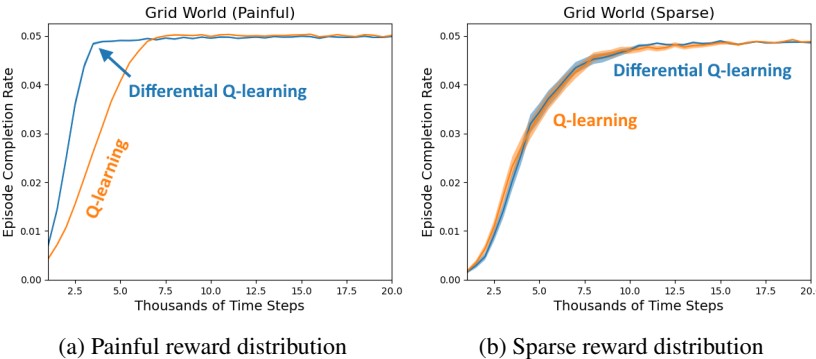

(a) Painful reward distribution         (b) Sparse reward distribution

Figure 1: Performance of Q-learning when used with reward- and value-centering compared against a standard uncentered baseline. The results are averaged over 100 independent runs where the shaded areas (occasionally less than a line width) represent the standard error.

Figure 1 shows the average rate of completed episodes per environment step of each algorithm's best parameter setting in terms of total episodes completed, for each reward distribution. With the painful reward distribution, differential Q-learning improves significantly over the uncentered baseline. However, in the sparse reward variant, both algorithms performed similarly. Recognizing that both algorithms performed worse with sparse rewards, it is possible that learning was bottlenecked by having a comparatively difficult exploration problem. Nevertheless, this validates that

there is benefit to centering in episodic problems. The results further suggest that the benefit can be expected when there is greater value-deviation across states (typical of dense reward settings), consistent with the intuition of centering reducing the distance that outputs need to travel. The value deviation along the optimal path is visualized in Figure 2.

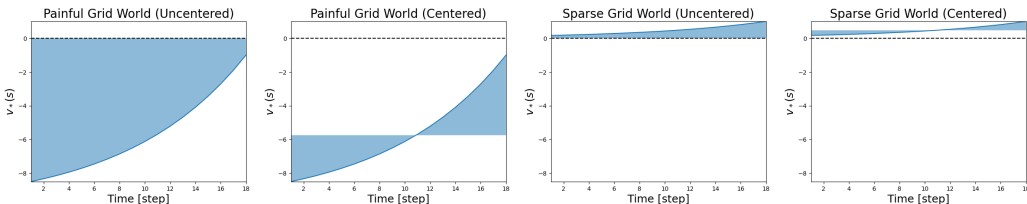

Figure 2: Uncentered and centered output distances (shaded area) along the optimal path in the Painful and Sparse Grid World environments. Because the Sparse Grid World's values are relatively small and concentrated around zero, we might expect less benefit from centering.

To further validate that centering can be done in episodic problems without changing the underlying problem and to demonstrate that there is benefit in doing so, we examine more challenging environments that require non-linear function approximation. Specifically, we extend two streaming deep reinforcement learning algorithms: Stream Q($\lambda$) and Stream AC($\lambda$) (Elsayed et al., 2024) to their differential counterparts. We additionally compare against PopArt (van Hasselt et al., 2016)—an algorithm which similarly employs output centering but with explicit attention on precisely preserving the unnormalized outputs. When using PopArt normalization with Stream Q($\lambda$) or Stream AC($\lambda$), we omit reward scaling as PopArt performs its own output scaling. We provide further experimental details and hyperparameters used in Appendix B.

Figure 3 shows the performance of Stream Q($\lambda$), differential Stream Q($\lambda$), and PopArt Stream Q($\lambda$) on Asterix, Breakout, Freeway, Seaquest, and SpaceInvaders from the MinAtar suite (Young & Tian, 2019). We tune $\eta$ and present results under the best-performing parameters from our search. We observe that differential Stream Q($\lambda$) improves over its uncentered base algorithm in all environments except for Breakout, where they perform similarly. On the other hand, PopArt normalization with Stream Q($\lambda$) was less consistent across the environments. It is unclear why this is the case because PopArt had not previously been demonstrated in a streaming deep reinforcement learning setup, and had not been used in these environments. It may be a nuance around explicitly normalizing the outputs with specific statistics and trying to precisely preserve outputs as these statistics may shift, in contrast with differential TD which jointly optimizes the shift under the same objective.

Next, we compare our centering approach in continuous-action control. In Figure 4, we show the performance of Stream AC($\lambda$), differential Stream AC($\lambda$), and PopArt Stream AC($\lambda$) in the MuJoCo suite (Todorov et al., 2012). It can be observed in Figure 4 that differential Stream AC($\lambda$) showed considerable improvement in the Ant-v4 and HalfCheetah-v4 environments, while not performing worse than its uncentered counterpart in the remaining ones. Notably, these two environments saw the largest return magnitudes over the duration of a run, which may be related to large value deviations across states. PopArt did not demonstrate statistically significant improvement in this suite.

Lastly, to explicitly validate the insight from the grid world experiments on when differential TD helps, we modified the Deepmind Control Suite's Reacher environment Tassa et al. (2018). Specifically, we created a *Painful* Reacher environment that receives a reward of $-1$ per step to mirror the grid world set up that showed substantial benefit. To lengthen episode duration and consequently increase value magnitudes and deviation, we additionally evaluate in a harder variant of the task that shrinks the goal location. With results presented in Figure 5, we see significant improvement in using differential Stream AC($\lambda$) over the uncentered base algorithm. Taking all of the evaluation together, we have established that reward centering can be done in episodic problems and that it can improve sample efficiency over uncentered algorithms. We further observed that the differential extension, when tuned, never performed worse than its base algorithm. This is to be expected because the $\eta = 0$ extreme results in a standard, uncentered TD update.

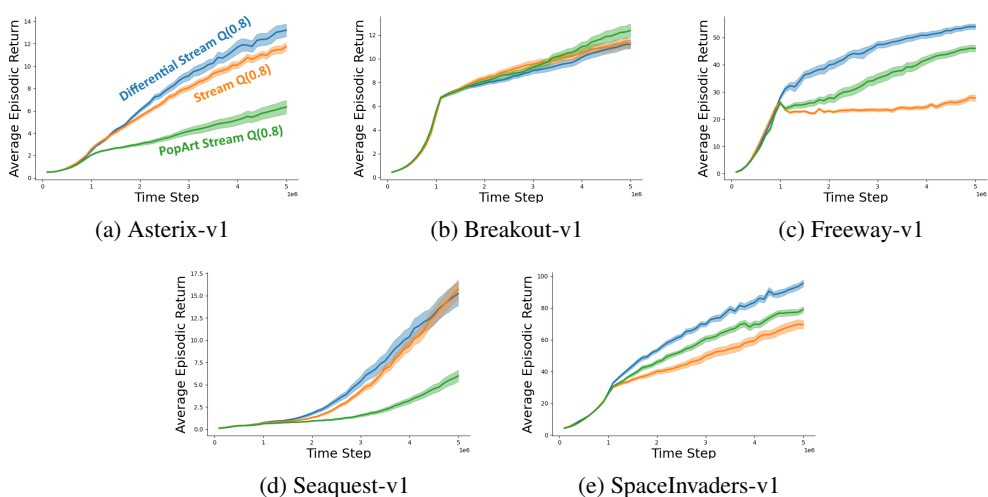

Figure 3: Performance of differential Stream Q(0.8) compared against a standard uncentered baseline and a PopArt-normalized baseline in the MinAtar suite. The results are averaged over 30 independent runs where the shaded areas represent the standard error.

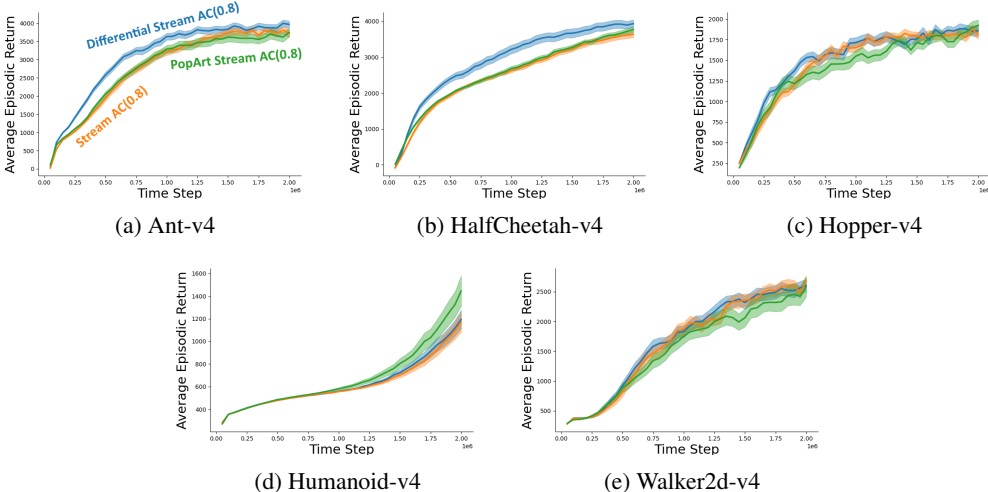

Figure 4: Performance of differential Stream AC(0.8) compared against a standard uncentered baseline and a PopArt-normalized baseline in the MuJoCo suite. The results are averaged over 30 independent runs where the shaded areas represent the standard error.

# 6 CONCLUSIONS, DISCUSSION, AND FUTURE WORK

In this work, we explored the reward centering mechanism of differential TD algorithms, which was previously limited to infinite-horizon reinforcement learning problems. By viewing reward centering from the lens of potential-based reward shaping, we propose a differential terminal value definition which—when used—maintains the ordering of policies and strictly generalizes differential TD to be applicable in episodic problems. We further show equivalence between the generalized differential TD update and an output-level, state- and action-independent bias unit. This establishes that the algorithm shares the theoretical guarantees previously shown for linear TD, and provides insight into how the centering term can be interpreted in an episodic problem. In a tabular environment, we demonstrated that centering can improve sample efficiency in episodic problems and provided arguments for when such benefits might be expected. In a streaming deep reinforcement learning setup, we further showed that these algorithms can scale to difficult problems with non-linear func-

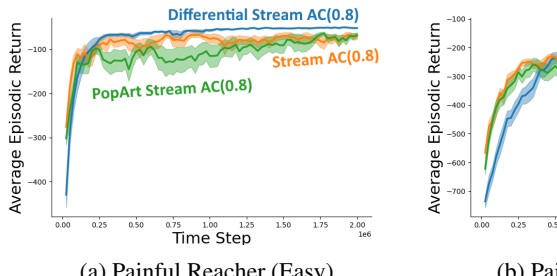

(a) Painful Reacher (Easy)     (b) Painful Reacher (Hard)

Figure 5: Performance of differential Stream AC(0.8) compared against a standard uncentered baseline and a PopArt-normalized baseline in the Painful Reacher environment. The results are averaged over 30 independent runs where the shaded areas represent the standard error.

tion approximation. Altogether, we have shown that reward centering can be applied in the presence of termination without altering the underlying task, and that doing so is beneficial.

There are many avenues for future work. Our evaluation focused on the streaming reinforcement learning setting, as that is where normalization was recently shown to have substantial benefit. However—as emphasized by Naik et al. (2024)—reward centering is a relatively general idea that can be easily dropped into any existing algorithm. Broadening its applicability toward episodic environments, the scope of possible comparisons between algorithms is larger now and there is merit in investigating differential TD's utility toward other types of episodic reinforcement learning algorithms (e.g., ones which store and process explicit episode trajectories). While the additional step-size parameter $\eta$ was already present in the original differential TD algorithms, the additional overhead in tuning this parameter remains a limitation. Given that centering involves learning a single scalar—a seemingly simple learning problem—it would be promising to explore whether $\eta$ can be efficiently meta-learned (e.g., Sutton, 1992; Mahmood et al., 2012; Sharifnassab et al., 2024).

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

## A    EXPERIMENTAL DETAILS OF GRID WORLD EXPERIMENTS

We swept over $\alpha \in \{0.1, 0.2, 0.3, 0.4, 0.5, 0.6, 0.7, 0.8, 0.9, 1.0\}$ for both algorithms and $\eta \in \{10^{-4}, 10^{-3.5}, 10^{-3}, 10^{-2.5}, 10^{-2}, 10^{-1.5}, 10^{-1}, 10^{-0.5}, 10^{0}\}$ for differential Q-learning. In the painful grid world, $\alpha = 1.0$ was best for both algorithms, with $\eta = 10^{-3}$ performing best for differential Q-learning. In the sparse grid world, $\alpha = 0.9$ was best for both algorithms, with $\eta = 10^{-4}$ performing best for differential Q-learning.

## B    EXPERIMENTAL DETAILS OF STREAMING DEEP RL EXPERIMENTS

We swept over bias step-sizes $\eta \in \{10^{-2}, 10^{-1}, 10^{0}, 10^{1}, 10^{2}, 10^{3}\}$, and we show performance with the best-performing value. For Stream AC, we used step-size $\alpha = 1$, $\kappa_{\pi} = 3$, $\kappa_v = 2$, $\lambda = 0.8$, discount factor $\gamma = 0.99$, and entropy coefficient $\tau = 0.01$. For Stream Q, we used step-size $\alpha = 1$, $\kappa_v = 2$, $\lambda = 0.8$, discount factor $\gamma = 0.99$. We used the same neural network architectures used with Stream AC and Stream Q reported by Elsayed et al. (2024). Lastly, we used an $\epsilon$-greedy policy where $\epsilon$ linearly decayed from 1 to 0.01 within 20% of the total time steps of a run.

## C    $b$'s INTERPRETATION IN AN EPISODIC PROBLEM

Prior average reward definitions lead to zero average reward in episodic problems (due to the equivalence between terminal states and an infinite loop of zero reward), that it is more informative to consider the bias-unit perspective in understanding what $b$ tends toward. Under a squared loss, the minimizing bias is the expectation of the targets under the behavior distribution. However, the bias is applied on a *reward level*, suggesting that $b$ is related to the expected state-value, but subdivided over the time remaining in an episode and discounted appropriately:

$$\mathbb{E}_{s \sim d_{\pi}}\left[\sum_{k=0}^{T-t-1} \gamma^k (R_{t+k+1} - b)\middle| s = S_t\right] = 0$$

$$\mathbb{E}_{s \sim d_{\pi}}\left[\sum_{k=0}^{T-t-1} \gamma^k b \middle| s = S_t\right] = \mathbb{E}_{s \sim d_{\pi}}\left[\sum_{k=0}^{T-t-1} \gamma^k R_{t+k+1}\middle| s = S_t\right]$$

$$\mathbb{E}_{s \sim d_{\pi}}\left[b \frac{1 - \gamma^{T-t}}{1 - \gamma}\middle| s = S_t\right] = \mathbb{E}_{s \sim d_{\pi}}[v_{\pi}(s)]$$

$$\mathbb{E}_{s \sim d_{\pi}}\left[b \frac{1 - \gamma^{T(s)}}{1 - \gamma}\right] = \mathbb{E}_{s \sim d_{\pi}}[v_{\pi}(s)]$$

$$b = \mathbb{E}_{s \sim d_{\pi}}\left[v_{\pi}(s) \frac{1 - \gamma}{1 - \gamma^{T(s)}}\right]$$

where $d_{\pi}$ represents normalized expected state visitation counts over non-terminal states under policy $\pi$ and $T(s)$ is the expected remaining episode length from state $s$.

## D    EPISODIC PROBLEMS AS STATE-DEPENDENT DISCOUNTING

It has been previously acknowledged that episodic problems can be implemented as infinite-horizon problems with a state-dependent discount function (Sutton, 1995; Sutton et al., 2011; White, 2016). For example, we can have $\gamma(s') = 0$ if $s'$ is terminal, and have it equal to the problem's discount otherwise. The terminal state would then transition back to a start state.

Consider an infinite-horizon return with potential-based reward shaping and state-dependent discounting. We define $\gamma_t \stackrel{\text{def}}{=} \gamma(S_t)$ for notational convenience:

$$
\begin{aligned}
G_t^{\Phi} \stackrel{\text{def}}{=} & \sum_{k=t}^{\infty} \left( \prod_{i=t+1}^{k} \gamma_i \right) \left( R_{k+1} + F(S_k, A_k, S_{k+1}) \right) \\
= & \sum_{k=t}^{\infty} \left( \prod_{i=t+1}^{k} \gamma_i \right) \left( R_{k+1} + \gamma_{k+1} \Phi(S_{k+1}) - \Phi(S_k) \right) \\
= & \sum_{k=t}^{\infty} \left( \prod_{i=t+1}^{k} \gamma_i \right) R_{k+1} + \sum_{k=t}^{\infty} \left( \prod_{i=t+1}^{k} \gamma_i \right) \gamma_{k+1} \Phi(S_{k+1}) - \sum_{k=t}^{\infty} \left( \prod_{i=t+1}^{k} \gamma_i \right) \Phi(S_k) \\
= & \sum_{k=t}^{\infty} \left( \prod_{i=t+1}^{k} \gamma_i \right) R_{k+1} + \sum_{k=t+1}^{\infty} \left( \prod_{i=t+1}^{k} \gamma_i \right) \Phi(S_k) - \sum_{k=t+1}^{\infty} \left( \prod_{i=t+1}^{k} \gamma_i \right) \Phi(S_k) - \Phi(S_t) \\
= & \sum_{k=t}^{\infty} \left( \prod_{i=t+1}^{k} \gamma_i \right) R_{k+1} - \Phi(S_t)
\end{aligned}
$$

Due to the Markov property, the subtraction of $\Phi(S_t)$ will not impact the ordering of policies. This also highlights that the learned values are relative to the potential function (i.e., it is akin to initializing the value function to $\Phi(s)$). Let us now consider the following potential function:

$$
\Phi(s) \stackrel{\text{def}}{=} \frac{b}{1 - \gamma(s)}
$$

If we implement an episodic problem by defining $\gamma(S_T) \stackrel{\text{def}}{=} 0$ and modifying the transition dynamics such that terminal states transition to a starting state sampled from a starting state distribution (independent of action), there are three scenarios:

$$
F(s, a, s') = \begin{cases} -\frac{b}{1-\gamma(s)}, & \text{if } \gamma(s') = 0 \\ \gamma(s') \frac{b}{1-\gamma(s')} - b, & \text{if } \gamma(s) = 0 \\ \gamma(s') \frac{b}{1-\gamma(s')} - \frac{b}{1-\gamma(s)}, & \text{otherwise} \end{cases}
$$

If we assume that all non-zero discounts are constant (i.e., $\gamma_t = \gamma$), this simplifies to:

$$
F(s, a, s') = \begin{cases} -\frac{b}{1-\gamma}, & \text{if } \gamma(s') = 0 \\ \gamma \frac{b}{1-\gamma} - b, & \text{if } \gamma(s) = 0 \\ -b, & \text{otherwise} \end{cases}
$$

This resembles the result in Section 3, except we have an additional $\gamma \frac{b}{1-\gamma}$ term in the case where $\gamma(s) = 0$. This term is set up to cancel with a portion of the previous time step's $-\frac{b}{1-\gamma}$ term, leaving $-b$ behind. However, this case corresponds with transitioning *from* a terminal state. Because we do not typically learn values for terminal states, this target typically will not be used. The remaining scenarios are consistent with what we get from the explicit episodic return.

## E  CONVERGENCE OF EPISODIC DIFFERENTIAL TD

In this section, we analyze the asymptotic convergence of the Differential TD algorithm with discounting. We focus on linear function approximation, $V(s) = \phi(s)^{\top} w$, which subsumes the tabular case.

We adopt the Ordinary Differential Equation (ODE) method for stochastic approximation (Borkar, 2008). We first define the update rules and the expanded parameter space. Crucially, we utilize an *unrolled MDP* formulation to unify the analysis of continuing and episodic tasks.

### E.1 Update Rules and Expanded Features

The differential TD algorithm maintains a parameter vector $\boldsymbol{w}$ and a separate scalar bias estimate $b$ (related to the average reward or reward offset). The update for a transition $(S_t, A_t, R_{t+1}, S_{t+1})$ is given by:

$$\boldsymbol{w}_{t+1} = \boldsymbol{w}_t + \alpha_t \delta_t \boldsymbol{\phi}(S_t), \tag{2}$$

$$b_{t+1} = b_t + \eta \alpha_t \delta_t. \tag{3}$$

Here, $\alpha_t$ is the learning rate and $\eta > 0$ is a scalar multiplier for the bias learning rate. Based on Section 4, The TD error $\delta_t$ is defined as:

- **Continuing:** $\delta_t = R_{t+1} - (1-\gamma)b_t + \gamma\boldsymbol{\phi}(S_{t+1})^\top \boldsymbol{w}_t - \boldsymbol{\phi}(S_t)^\top \boldsymbol{w}_t$.
- **Episodic (Non-terminal):** Same as above.
- **Episodic (Terminal):** $\delta_t = R_{t+1} - b_t - \boldsymbol{\phi}(S_t)^\top \boldsymbol{w}_t$.

To analyze this coupled system, we augment the feature vector to include a bias unit, forming the expanded feature vector $\tilde{\boldsymbol{\phi}}(s) = [\mathbb{I}(s), \boldsymbol{\phi}(s)^\top]^\top \in \mathbb{R}^{d+1}$. The corresponding parameter vector is $\tilde{\boldsymbol{w}} = [b, \boldsymbol{w}^\top]^\top$. Here, $\mathbb{I}(s)$ acts as the bias feature: it is 1 for all states in the continuing setting, and $[\mathbf{e}]_s$ (indicator of non-terminal status) in the episodic setting.

The updates can be rewritten in a unified form:

$$\tilde{\boldsymbol{w}}_{t+1} = \tilde{\boldsymbol{w}}_t + \alpha_t \delta_t \boldsymbol{K} \tilde{\boldsymbol{\phi}}(S_t), \tag{4}$$

where $\boldsymbol{K} = \text{diag}(\eta, 1, \ldots, 1)$ handles the separate learning rate scaling, and the TD error simplifies to:

$$\delta_t = R_{t+1} + \gamma\tilde{\boldsymbol{\phi}}(S_{t+1})^\top \tilde{\boldsymbol{w}}_t - \tilde{\boldsymbol{\phi}}(S_t)^\top \tilde{\boldsymbol{w}}_t. \tag{5}$$

In the episodic case, we define $\tilde{\boldsymbol{\phi}}(S_{\text{term}}) \equiv \boldsymbol{0}$.

### E.2 Assumptions and Unrolled MDP

To provide a single convergence proof, we model the episodic setting as a continuing process and formalize our assumptions.

**Definition 1** (Unrolled MDP). *For an episodic task, the Unrolled MDP is a continuing Markov chain constructed by treating the sequence of episodes as a single stream. Upon reaching a terminal state $s_T$, the process transitions at the next time step to a start state $s_0$ sampled from the initial distribution $d_0$.*

We define the unified transition matrix $\boldsymbol{P}_\pi$ and stationary distribution matrix $\boldsymbol{D}_\pi$ (diagonal matrix of the stationary distribution $d_\pi$) based on this unrolled view.

**Assumption 1** (Ergodicity). *The Markov chain induced by the policy $\pi$ (or the Unrolled MDP in the episodic case) is ergodic (irreducible and aperiodic), admitting a unique stationary distribution $d_\pi$.*

**Assumption 2** (Linearly Independent Features). *The expanded feature matrix $\tilde{\boldsymbol{\Phi}}$ has full column rank.*

- *Continuing: $\tilde{\boldsymbol{\Phi}} = [\mathbf{1}, \boldsymbol{\Phi}]$.*

- *Episodic: $\tilde{\boldsymbol{\Phi}} = [\mathbf{e}, \boldsymbol{\Phi}]$, where $\mathbf{e}$ is zero for terminal states.*

*Furthermore, for all $s$, $\|\boldsymbol{\phi}(s)\| < \infty$.*

**Assumption 3** (Step Sizes and Noise). *The step sizes $\alpha_t$ satisfy the standard Robbins-Monro conditions: $\sum_{t=0}^\infty \alpha_t = \infty$ and $\sum_{t=0}^\infty \alpha_t^2 < \infty$. The reward function has bounded variance.*

### E.3 Convergence Analysis

The behavior of the stochastic update in Equation 4 is governed by the mean field ODE:

$$\dot{\tilde{\boldsymbol{w}}} = \boldsymbol{K}(\tilde{\boldsymbol{A}}\tilde{\boldsymbol{w}} + \tilde{\boldsymbol{b}}), \tag{6}$$

where $\tilde{\boldsymbol{b}} = \mathbb{E}[R_{t+1}\tilde{\boldsymbol{\phi}}(S_t)]$ and $\tilde{\boldsymbol{A}}$ is the expected update direction matrix:

$$\tilde{\boldsymbol{A}} = \tilde{\boldsymbol{\Phi}}^\top \boldsymbol{D}_\pi (\gamma \boldsymbol{P}_\pi - \boldsymbol{I})\tilde{\boldsymbol{\Phi}}. \tag{7}$$

**Theorem 1.** *Let $\gamma < 1$ and $\eta > 0$. Under Assumptions 1, 2, and 3, the parameter $\tilde{\boldsymbol{w}}_t$ converges with probability 1 to the unique fixed point $\tilde{\boldsymbol{w}}^* = -\tilde{\boldsymbol{A}}^{-1}\tilde{\boldsymbol{b}}$.*

*Proof.* The proof proceeds in two steps. First, we show that $\tilde{\boldsymbol{A}}$ is negative definite. Second, we show that the preconditioning by $\boldsymbol{K}$ preserves stability.

**Step 1: Negative Definiteness of $\tilde{\boldsymbol{A}}$.** Consider the quadratic form for an arbitrary vector $\mathbf{x} \neq \mathbf{0}$. Let $\mathbf{y} = \tilde{\boldsymbol{\Phi}}\mathbf{x}$. By Assumption 2 (full rank), $\mathbf{y} \neq \mathbf{0}$.

$$\mathbf{x}^\top \tilde{\boldsymbol{A}}\mathbf{x} = \mathbf{y}^\top \boldsymbol{D}_\pi (\gamma \boldsymbol{P}_\pi - \boldsymbol{I})\mathbf{y} = \gamma\langle \mathbf{y}, \boldsymbol{P}_\pi \mathbf{y}\rangle_{\boldsymbol{D}_\pi} - \|\mathbf{y}\|_{\boldsymbol{D}_\pi}^2. \tag{8}$$

The transition operator is a non-expansion in the $\boldsymbol{D}_\pi$-weighted norm (Tsitsiklis & Van Roy, 1997). Applying the Cauchy-Schwarz inequality:

$$\langle \mathbf{y}, \boldsymbol{P}_\pi \mathbf{y}\rangle_{\boldsymbol{D}_\pi} \leq \|\mathbf{y}\|_{\boldsymbol{D}_\pi}\|\boldsymbol{P}_\pi \mathbf{y}\|_{\boldsymbol{D}_\pi} \leq \|\mathbf{y}\|_{\boldsymbol{D}_\pi}^2. \tag{9}$$

Substituting this back yields:

$$\mathbf{x}^\top \tilde{\boldsymbol{A}}\mathbf{x} \leq (\gamma - 1)\|\mathbf{y}\|_{\boldsymbol{D}_\pi}^2. \tag{10}$$

Since $\gamma < 1$, the quadratic form is strictly negative. Thus $\tilde{\boldsymbol{A}}$ is negative definite (and consequently Hurwitz).

**Step 2: Stability with $\eta > 0$.** The system matrix of the ODE is $\tilde{\boldsymbol{A}}_\eta = \boldsymbol{K}\tilde{\boldsymbol{A}}$. We analyze its spectrum via a similarity transformation. Consider $\boldsymbol{S} = \boldsymbol{K}^{-1/2}\tilde{\boldsymbol{A}}_\eta \boldsymbol{K}^{1/2}$. Substituting $\tilde{\boldsymbol{A}}_\eta = \boldsymbol{K}\tilde{\boldsymbol{A}}$, we obtain:

$$\boldsymbol{S} = \boldsymbol{K}^{-1/2}(\boldsymbol{K}\tilde{\boldsymbol{A}})\boldsymbol{K}^{1/2} = \boldsymbol{K}^{1/2}\tilde{\boldsymbol{A}}\boldsymbol{K}^{1/2}. \tag{11}$$

We examine the definiteness of $\boldsymbol{S}$. For any non-zero vector $\mathbf{u}$:

$$\mathbf{u}^\top \boldsymbol{S}\mathbf{u} = \mathbf{u}^\top \boldsymbol{K}^{1/2}\tilde{\boldsymbol{A}}\boldsymbol{K}^{1/2}\mathbf{u}. \tag{12}$$

Let $\mathbf{v} = \boldsymbol{K}^{1/2}\mathbf{u}$. Since $\eta > 0$, $\boldsymbol{K}$ is positive definite, implying $\mathbf{v} \neq \mathbf{0}$. The expression simplifies to $\mathbf{v}^\top \tilde{\boldsymbol{A}}\mathbf{v}$. From Step 1, we know $\mathbf{v}^\top \tilde{\boldsymbol{A}}\mathbf{v} < 0$. Therefore, $\mathbf{u}^\top \boldsymbol{S}\mathbf{u} < 0$, meaning $\boldsymbol{S}$ is negative definite.

Since $\boldsymbol{S}$ is negative definite, all its eigenvalues have strictly negative real parts. Because $\tilde{\boldsymbol{A}}_\eta$ is similar to $\boldsymbol{S}$, they share the same eigenvalues. Thus, $\tilde{\boldsymbol{A}}_\eta$ is Hurwitz. By standard stochastic approximation theory (Borkar, 2008), the iteration converges globally to the unique fixed point $\tilde{\boldsymbol{w}}^*$. $\qquad\square$

