# OpenReview forum: "Extending Differential Temporal Difference Methods to Episodic Problems"
_ICLR.cc/2026/Conference — Submitted to ICLR 2026_

### Official Review · Reviewer_QdqX · 2025-10-25

**Soundness:** 1
**Presentation:** 2
**Contribution:** 1
**Rating:** 2
**Confidence:** 3

**Summary:**

Differential temporal difference (TD) methods, which rely on reward centering via the average reward, are generally inapplicable to episodic problems because centering can alter the optimal policy. This work proposes a generalization of differential TD to overcome this limitation, making it applicable to both discounted and undiscounted episodic contexts. The core mechanism involves defining a differential terminal value and proving that the modification preserves optimal policy invariance by mapping the generalized update onto the framework of potential-based reward shaping. The authors also establish a theoretical equivalence between the algorithm and a form of linear TD with a state-action-independent output-level bias unit. Empirical results in streaming deep RL environments (MinAtar, MuJoCo) suggest that the proposed generalization, integrated into Stream Q($\lambda$) and Stream AC($\lambda$), improves sample efficiency compared to uncentered baselines.

**Strengths:**

- The writing in this paper is clear, and I can easily grasp the information the authors intend to convey.
- The performance experiments in the paper demonstrate that the method has some effectiveness.

**Weaknesses:**

- I believe the novelty of this paper is very limited; it merely incorporates commonly used terms from reward shaping methods into differential Q Learning, without offering any new innovation.
- The paper does not provide any theoretical or experimental analysis that offers insight; the experiments are almost entirely performance evaluations.
- The amount of work presented in this paper is modest, and I believe the contribution is not sufficient to meet the bar for ICLR.

**Questions:**

- Could you provide some valuable insights through theoretical or experimental analysis to justify the necessity of the method proposed in the paper?

---

> ### Author Response · Authors · 2025-11-17
> **Response**
>
> > I believe the novelty of this paper is very limited; it merely incorporates commonly used terms from reward shaping methods into differential Q Learning, without offering any new innovation.
>
> We note that not only has Differential TD not been used in episodic problems previously, but the literature repeatedly, explicitly states that it is not applicable to such problems as if a fundamental limitation (e.g., Naik et al., 2024; Lee et al., 2025).
>
> > Could you provide some valuable insights through theoretical or experimental analysis to justify the necessity of the method proposed in the paper?
>
> Certainly. Below are, in our opinion, insights the paper provides:
>
> * Differential TD *can* be applied to episodic problems while maintaining the *complete ordering of policies* via a terminal value definition. As mentioned above, prior work on Differential TD has mentioned its inapplicability to episodic problems as a fundamental or theoretical limitation, e.g., in terms of soundness or guarantee of optimizing the same objective. We believe the simplicity of the extension is a plus, and believe it warrants such attention as Differential TD is often not considered at all due to its roots in the average reward setting. Reward centering by Naik et al. (2024) helped broaden Differential TD beyond average reward objectives by showing that it can also more effectively optimize discounted objectives, but it remained constrained to infinite-horizon problems. This work broadens it to both discounted and undiscounted, episodic and infinite-horizon problems (including finite-horizon settings).
>
> * The equivalence between Differential TD and a state-action-independent bias unit. Prior work has actually stated that it "may seem similar to, but is different from" a state-action-independent bias unit (Naik et al., 2024). We show that under the usual Differential TD update, it is *exactly* equivalent to this, and by extension, inherits theoretical guarantees of linear TD. We also emphasize that this is not just a bias unit in the final hidden layer of a neural network, as each output would have a separate weight for it and credit assignment would based on how often an action is selected. While it might seem like a trivial application of potential-based reward shaping, this is a dynamically-varying potential function which under different update rules, could be non-convergent (e.g., periodic oscillation between the shaping term and the relative values).
>
> * The empirical evaluation shows that not only can Differential TD be sound in episodic problems, it can improve sample efficiency of learning value functions. This applies to both pure value-based methods as well as the critic in an actor-critic setup. Notably, there are key intuitions around when it is expected to help (large value deviation over states), and has trivial parameter settings which guarantee that it never does worse.
>
> * A nuance of soft- and hard-termination is that it gives semantics to zero (e.g., discounting tends values to zero), which has interplay with how value functions are initialized, optimism, etc. Centering decouples termination from enforcing semantics around zero, as termination now tends toward the value function's mean over the visitation distribution (or any specifiable reference, in the case of shifting but not centering).
>
> > the experiments are almost entirely performance evaluations.
>
> We believe that the performance evaluations are in line with the claims we were trying to make (e.g., centering is akin to initializing the value function to its mean, thereby speeding up learning). We have some additional empirical results validating that the optimal policy is preserved, which we felt did not add much as it is clear that this is the case from the reward shaping perspective. We further have results showing faster policy evaluation (i.e., centered values can be learned faster than uncentered ones, and can be un-shifted if needed), but these are similarly a performance evaluation. We can include these results in the appendices of our revision, but along these lines, we'd like to ask what experiments the reviewer would like to see to strengthen the paper.
>
> > The amount of work presented in this paper is modest, and I believe the contribution is not sufficient to meet the bar for ICLR
>
> We respectfully disagree—our work fills a gap in the literature that has prevented researchers from using reward centering in episodic settings (as explicitly noted in Lee et al., 2025). While our method is simple, the underlying idea and variety of perspectives on it are non-trivial and leads to clear empirical benefits. We would appreciate any actionable feedback on what the reviewer believes would strengthen the paper to meet the acceptance bar.
>
> ## References
>
> * Naik, A., Wan, Y., Tomar, M., Sutton, R. S. (2024). Reward centering.
> * Lee, H., Lee, Y., Seno, T., Kim, D., Stone, P., Choo, J. (2025). Hyperspherical normalization for scalable deep reinforcement learning.

---

> > ### Comment · Reviewer_QdqX · 2025-11-27
> >
> > I thank the authors for their response. However, while the authors provided a verbal explanation, they failed to present sufficient arguments to establish the necessity of the proposed method in an episodic setting.
> >
> > A more convincing approach would be to articulate this through formal theorems accompanied by proofs; unfortunately, the original manuscript lacks such theorems, and no revisions were made to the paper during the rebuttal phase.
> >
> > Furthermore, given that the paper targets the episodic setting, the most compelling evidence would come from demonstrating significant advantages in episodic tasks, such as robotic manipulation. In the reported experiments (on simple tasks like Gridworld and MuJoCo), the differential versions of Q-Learning/Stream Q/Stream AC show no distinct advantage over their vanilla counterparts, as the final converged rewards are evidently very similar.
> >
> > In summary, I remain skeptical regarding the effectiveness of the proposed method and its contribution to the field of reinforcement learning.

---

> > > ### Author Response · Authors · 2025-12-03
> > > **Response**
> > >
> > > We thank the reviewer for their additional feedback. While the discussion can't proceed further due to the circumstances around this conference, we want to highlight that we have added a complete proof of convergence in the appendix of our revision. We apologize for not posting a revision sooner—this was in part from details being worked out, along with having not received further feedback to our rebuttal until your latest response.
> > >
> > > Regarding the supposed lack of a distinct advantage over vanilla counterparts, final performance is only one dimension to consider improvements in. It is clear (with statistical significance over 30 independent runs) that in many cases, sample efficiency was improved. From negligible overhead, this translates directly to improved compute efficiency and wall-clock time for the same performance. This is particularly relevant to the streaming setting that treats every moment equally, as opposed to explicit, separate data collection/storage and batch-processing (e.g., PPO). We hope that this clarifies things and is more convincing, at least in the context of what we claim in the paper.

---

### Official Review · Reviewer_rykv · 2025-10-28

**Soundness:** 3
**Presentation:** 3
**Contribution:** 3
**Rating:** 8
**Confidence:** 3

**Summary:**

This paper revisits differential temporal difference (TD) learning, a class of value-based reinforcement learning (RL) algorithms traditionally applied to continuing infinite-horizon problems. Differential TD methods rely on reward centering, subtracting the average reward from each observed reward to stabilize learning and remove constant value offsets. However, this reward transformation is not invariant to optimal policies in episodic tasks, because subtracting a constant shifts the value of trajectories of different lengths unequally. As a result, differential TD could distort the policy ranking in problems with termination.
The authors propose a generalization of differential TD that extends its applicability to episodic reinforcement learning, both discounted and undiscounted. By reinterpreting reward centering through the framework of potential-based reward shaping, they show that adding a specific terminal correction preserves the ordering of optimal policies in episodic environments. Moreover, they demonstrate the equivalence between the proposed generalization and a TD update with an explicit bias unit, a state- and action-independent output parameter that functions as a learnable baseline. This equivalence allows the generalized differential TD to inherit theoretical guarantees from linear TD methods.
Overall, the work provides both theoretical and empirical evidence that reward centering can be safely and beneficially applied in episodic reinforcement learning.

**Strengths:**

The central contribution, viewing differential TD as a special case of potential-based reward shaping with terminal correction, is elegant and rigorous. The analysis in Sections 3–4 clearly shows how to define a terminal differential value so that subtracting the average reward does not alter the policy ranking. The reinterpretation of differential TD as TD with a bias unit (Eq. 1–5) is particularly insightful, bridging a gap between continuing and episodic settings.
The equivalence proof allows the algorithm to inherit established convergence properties of linear TD, giving the proposal strong theoretical grounding.
The paper balances formal reasoning with practical implications. For example, the grid world examples (Fig. 1–2) are simple yet clearly demonstrate when and why centering helps (notably under “painful” dense reward conditions). The potential-based reward shaping analysis (Section 3) naturally connects to the algorithmic implementation (Algorithm 1) and the bias-unit interpretation (Section 4). The theoretical argument that episodic problems can be viewed as continuing problems with state-dependent discounting (Appendix D) is a clean conceptual unification.
The authors evaluate across diverse scales and settings. In almost all settings, differential versions improved or matched baseline performance, while never performing worse, supporting the claim that reward centering is safe and often beneficial.
The proposed changes are minimal and easily applicable: Only one additional scalar parameter (b) and one step-size are introduced. The approach can be applied to existing algorithms “as-is,” requiring no structural change. This makes it highly practical for resource-constrained or streaming RL settings, where buffer-free online learning is essential.
Appendices A–D list exact hyperparameters, search ranges, and algorithmic details.
The pseudocode (Algorithm 1) is clear and unambiguous. The paper also promises to release source code upon acceptance.

**Weaknesses:**

While the reinterpretation is elegant, it largely builds on existing principles (potential-based reward shaping, bias-augmented TD, and differential TD) rather than introducing fundamentally new algorithms or analyses.
The main theoretical novelty lies in recognizing how to apply these concepts to episodic settings, but the resulting mathematics is straightforward.
The tabular experiment (Figure 1) is limited to two simple grid worlds.
Although the results serve their illustrative purpose, including more challenging episodic tasks (e.g., stochastic transitions or larger state spaces) could provide stronger validation.
The additional step-size parameter (eta) introduces a hyperparameter tuning burden.
While this is acknowledged (Sec. 6), more analysis on its sensitivity and its relationship to alpha or gamma would improve the reader’s understanding

**Questions:**

1.	Clarify the theoretical connection between Section 3 and 4.
While the potential-based shaping and bias-unit views are both valid, a more direct derivation linking the two would help readers see how the bias term implements the shaping correction in function approximation.
2.	Add computational analysis.
While the method introduces negligible overhead, including runtime or memory comparisons would concretely support claims of “minimal cost.”
3.	Extend to policy-based methods.
Since reward centering parallels baseline subtraction in actor-critic methods, exploring whether similar bias-units could be applied to the critic in PPO or A2C would be a natural extension.

---

> ### Author Response · Authors · 2025-11-17
> **Response**
>
> > Clarify the theoretical connection between Section 3 and 4. While the potential-based shaping and bias-unit views are both valid, a more direct derivation linking the two would help readers see how the bias term implements the shaping correction in function approximation.
>
> Interestingly, the derivation linking the two is to follow the steps in Section 4 in reverse. Section 4 ended with an update that can be interpreted as having the aforementioned reward shaping term, along with an update rule for it. The role of section 3 is to show that the terminal value redefinition, for any value of $b$, will ensure the underlying problem does not change. We will clarify more the role each section plays, and how they are related, in the main text.
>
> > Add computational analysis. While the method introduces negligible overhead, including runtime or memory comparisons would concretely support claims of “minimal cost.”
>
> The memory overhead is an additional float, and the computation involves an additional subtraction in the TD error with an additional update $b = b + \eta\alpha\delta$. In terms of wall-clock time, there was practically no noticeable difference as the computation of performing forward and backward passes through a neural network are several orders of magnitude larger than the 5 extra simple operations involved with reward centering. Using eligibility traces roughly doubles the memory and computation, but it remains negligible (2 scalars and around 10 simple operations). Of note, because the additional scalar is state-action independent, the additional computation is fixed and does not get more intensive as the function approximation complexity grows.
>
> > Extend to policy-based methods. Since reward centering parallels baseline subtraction in actor-critic methods, exploring whether similar bias-units could be applied to the critic in PPO or A2C would be a natural extension.
>
> We'd like to note that this is actually included in the paper through extending Streaming Actor-Critic($\lambda$) (Elsayed et al., 2024) to its differential version. This was done as you have described, where the critic used differential TD updates.
>
> ## References
>
> * Elsayed, M., Vasan, G., Mahmood, A. R. (2024). Streaming deep reinforcement learning finally works.

---

### Official Review · Reviewer_Sedk · 2025-10-31

**Soundness:** 2
**Presentation:** 3
**Contribution:** 1
**Rating:** 2
**Confidence:** 3

**Summary:**

The paper proposes a modification of differential TD methods based on reward centering in episodic problems. First, the authors provide the modification in the differential TD algorithm in the presence of the terminal state. Further modification in the learned loss is done. Finally, the authors evaluate their proposed differential TD algorithm (under episodic reward scenarios) in the tabular setting and Atari and Mujoco scenarios with the integration of the actor-critic algorithm. Experiment results show that the efficacy of reward centering is prevalent in the 'painful' reward setting, where the reward is not normalized.

**Strengths:**

1) The motivation to use reward centering makes sense. Reward centering can facilitate fast adaptation of the base RL algorithm under unnormalized rewards.

2) The paper is not hard to follow. I believe the authors thoroughly extended the reward centering to the condition of episodic reward.

**Weaknesses:**

1) My biggest concern about the paper is its significance. The method can be efficient in some tailored scenarios, but the gap becomes low in the sparse reward scenario. Furthermore, there are myriad of methods that extend over Q-Learning and the AC algorithm. However, the paper only includes Q-Learning and actor-critic, which can be seen as outdated. More relatively recent methods [1,2,3] should also be compared.

2) I am not sure of the contribution of the paper. The author's work extends differential TD learning from an infinite-horizon scenario to the scenario where there terminal state exists, which feels like a straightforward extension. Neither does the significance of the result, connected to 1).

3) I also do not understand the motivation for extending differential TD learning methods to episodic tasks. Reward centering can be effective in no non-terminal state. However, if the process have terminal state, what is the efficacy of using complex reward centering instead of advantage function?

4) Enhanced exploration method [4] or efficient update through experience replay [5] also greatly contributes to the sample-efficient update of reinforcement learning methods. The gain of the proposed episodic version can be diminished under the enhanced exploration or replay-based update.

***References***

[1] Soft actor-critic: Off-policy maximum entropy deep reinforcement learning with a stochastic actor, ICML 2018

[2]  Rainbow: Combining improvements in deep reinforcement learning, AAAI 2018.

[3] Asynchronous Methods for Deep Reinforcement Learning, ICML 2016.

[4] Exploration by Random Network Distillation, ICLR 2019.

[5] Revisiting Fundamentals of Experience Replay, ICML 2020

**Questions:**

See Weaknesses

---

> ### Author Response · Authors · 2025-11-17
> **Response**
>
> > My biggest concern about the paper is its significance. The method can be efficient in some tailored scenarios, but the gap becomes low in the sparse reward scenario. Furthermore, there are myriad of methods that extend over Q-Learning and the AC algorithm. However, the paper only includes Q-Learning and actor-critic, which can be seen as outdated. More relatively recent methods [1,2,3] should also be compared.
>
> We understand the confusion but would like to highlight that we specifically extended and compared with *Streaming* Q($\lambda$) and *Streaming* Actor-Critic($\lambda$) (Elsayed et al., 2024) which are *not the same* as Q-learning and AC and are *more recent* than any of the cited methods above. Notably, these streaming algorithms are what highlighted the critical role of normalization and were the primary motivation behind working out a sound way to incorporate reward centering into any problem setting. We further argue that it's a plus if there can be intuition regarding when a method can be expected to be efficient, especially when it can be set to never be worse.
>
> > I am not sure of the contribution of the paper. The author's work extends differential TD learning from an infinite-horizon scenario to the scenario where there terminal state exists, which feels like a straightforward extension. Neither does the significance of the result, connected to 1).
>
> While it may seem straight forward in that the return is already bounded, prior works have specifically stated that it *cannot be applied to episodic settings* due to the lack of optimal policy invariance, often phrasing it to imply that it's a fundamental or theoretical limitation (e.g., Naik et al., 2024). For example, Lee et al. (2025) explored many forms of normalization, but explicitly state that they "do not center the reward, as shifting the reward can alter the optimal policy in episodic tasks." Due to differential TD's roots in the average reward setting, such an extension has often been written off as just not being applicable in episodic problems.
>
> > I also do not understand the motivation for extending differential TD learning methods to episodic tasks. Reward centering can be effective in no non-terminal state. However, if the process have terminal state, what is the efficacy of using complex reward centering instead of advantage function?
>
> First, we argue that reward centering is *not* complex, as it introduces a single scalar parameter. When considering advantage functions, computing the advantage still requires learning a value function. Even if the goal is to learn uncentered values, based on our results as well as those by Naik et al. (2024), *it can still be quicker to learn centered values*, and then un-shift them by $\frac{b}{1 - \gamma}$ to get the uncentered values back. Therefore, even if one ultimately cares about advantage functions for a policy gradient method, more effective value learning directly translates to better estimates of the advantage, as demonstrated by the Streaming AC($\lambda$) results.
>
> > Enhanced exploration method [4] or efficient update through experience replay [5] also greatly contributes to the sample-efficient update of reinforcement learning methods. The gain of the proposed episodic version can be diminished under the enhanced exploration or replay-based update.
>
> These are not competing but are complementary directions and there is evidence that it is not diminished: Naik et al. (2024) has specifically demonstrated benefit of reward centering *with experience replay*. However, their work was limited to the infinite-horizon setting as again, differential TD as previously understood was not sound in episodic problems without influencing the underlying task (i.e., the ordering of policies). This work extends it to episodic problems while maintaining optimal policy invariance, and taken together with the results by Naik et al. (2024), it's expected that centering still complements—and is not diminished by—the mentioned techniques. The lack of emphasis on replay-based updates in our work is due to the specific motivation that normalization was shown to play a critical role in enabling streaming deep reinforcement learning (and centering is a form of normalization).
>
> ## References
> * Elsayed, M., Vasan, G., Mahmood, A. R. (2024). Streaming deep reinforcement learning finally works.
> * Naik, A., Wan, Y., Tomar, M., Sutton, R. S. (2024). Reward centering.
> * Lee, H., Lee, Y., Seno, T., Kim, D., Stone, P., Choo, J. (2025). Hyperspherical normalization for scalable deep reinforcement learning.

---

### Official Review · Reviewer_LPfW · 2025-11-01

**Soundness:** 3
**Presentation:** 2
**Contribution:** 2
**Rating:** 4
**Confidence:** 3

**Summary:**

The paper studies extension of differential TD-learning to episodic case. In the infintie horizon problem, average reward is often estimated and subtracated in the TD-learning udpate, which the authors call reward centering mechanism. This becomes a problem in episodic case as subtracting constant reward can cause early termination or non-terminating behavior. The invariance is achieved by considering a proper reward shaping at the terminal state. The paper approximates the value function with a differential value function and an additional bias unit which does not depend on state-action, and reflects the free variable in the reward shaping method.

**Strengths:**

1. The paper considers an unexplored problem in the community, which is average reward learning in episodic case. While the community has focused on the infinite horizon case, the paper considers a different scenario which is finite horizon case (episodic).

2. The authors manage to derive a new algorithm in average reward learning under episodic scenario. The derivation which relies on reward shaping seems to new.

2. The claims are supported by experimental results. The proposed method seems to be effective in the streaming learning scenario.

**Weaknesses:**

1. The authors do not propose a theoretical convergence result for their update equation.


2. The mean-squared formulation and the parametrization $V(s;w,b)=V^{\Delta}(s;w)+b\approx v^{\pi}(s)$ is not straightforward. Moreover, the motivation for the reparameterization of $\hat{b}=(1-\gamma) b$ and $\hat{\eta}=\eta(1-\gamma)$ is not clear.

3. skepticism on the problem formulation from theroetical perspective: The challenge of average reward problem is in its unboundedness of $\sum_{t=0}^{\infty} r_t$. Nonethelss, if we consider an episodic case, we always have bounded sum of rewards. With boundedness, we maybe possible to guarantee many nice properties as in the discounted infinite horizon problem. Therefore, it is questionable, whether considering an episdoic problem in the average reward is meaningful in theoretical sense.

**Questions:**

1. The intuition on introducing additional bias unit is not clear ($V^{\Delta}(s;w)+b$).
2. In line 242, can the authors provide more detail on the choice of $\eta$ and $b_0$?

---

> ### Author Response · Authors · 2025-11-17
> **Response**
>
> > ...do not propose a theoretical convergence result...
>
> The intent of section 4 is to demonstrate that Differential TD can be exactly interpreted as a specific choice of feature representation (an additional state-action-independent bias unit). The implications of this are that it can be directly mapped onto a form of linear TD, which is convergent. In other words, demonstrating this equivalence does establish a convergence result for the update rule. We apologize for this confusion and can make this more explicit in the main text.
>
> > ...mean-squared formulation and the parametrization is not straightforward... motivation for the reparameterization is not clear...  intuition on introducing additional bias unit is not clear...
>
> The mean-squared value-error formulation is the standard formulation in value-based RL (e.g., section 9.2 in Sutton \& Barto (2018), the loss in Mnih et al. (2015), etc.). where the goal is to learn the parameters $\mathbf{w}$ of a parameterized value function $V(s;\mathbf{w})\approx v_\pi(s)$.
>
> Our specific parameterization $V(s;\mathbf{w}, b) = V^\Delta(s;\mathbf{w}) + b$ is introduced as a definition, and the interpretation of it and what follows is that, if you define your value function to be a state-action independent scalar plus some residual, this is exactly equivalent to Differential TD's update with scaling factors $(1 -\gamma)$ on the hyperparameters (which can be absorbed into the choices of these parameters). As mentioned above, the motivation and purpose of this definition is to show that Differential TD is equivalent to a specific choice of linear feature representation, and establish that the update is convergent. We will clarify this motivation further in the main text.
>
> > ...skepticism on the problem formulation from theoretical perspective... we always have bounded sum of rewards... it is questionable, whether considering an episodic problem in the average reward is meaningful in theoretical sense.
>
> As mentioned in Appendix C, $b$ isn't best interpreted as average reward in this setting, as under the usual definition of the average reward, it would be zero in any episodic problem.
>
> Prior work on reward centering (Naik et al., 2024) established that under discounting, *where the return is already bounded*, there is still benefit to centering rewards. The theoretical benefit to it is illustrated in Figure 2, where it is akin to initializing the learned values to their mean, thereby reducing the distance that outputs need to travel (weighted by the visitation distribution). Because the mean is a relatively simpler learning problem, this generally speeds up learning for values with large deviations across states. Such centering is widely used in supervised learning due to favorable optimization properties, but doing it naively (i.e., without special care in handling terminal values) isn't sound in episodic reinforcement learning. For example, Lee et al. (2025) explored many forms of normalization, but explicitly states that they "do not center the reward, as shifting the reward can alter the optimal policy in episodic tasks." Our work addresses this limitation.
>
> > ...can the authors provide more detail on the choice of $\eta$ and $b_0$?
>
> If one were to introduce a state-action-independent bias unit to the value function parametrization, initialize $b_0$ arbitrarily, and choose some bias step size $\eta$, it is equivalent to using Differential TD with the bias unit and bias step size scaled by factors of $1 - \gamma$. To clarify, the choices in Section 4 were to prove equivalence, as the $1 - \gamma$ factors can get absorbed into the step size and initial bias (which in our work and the literature, is typically initialized to 0). To give a concrete example, if we run $\epsilon$-greedy Differential Q-learning with an initial $b = 0$, $\eta = 0.01$, it is exactly equivalent to using Q-learning with the value function parametrization $Q(s, a;\mathbf{w}, b) = Q^\Delta(s, a;\mathbf{w}) + b$ with $\eta = (1 - \gamma)0.01$ (and other parameters $\alpha$, $\epsilon$ kept the same). Here are quick runs for Differential Q-learning and Q-learning with a bias unit under the above parameter settings (and same seed) verifying that the subtracted term follows identical trajectories in a grid world: https://i.imgur.com/Pxhyfun.png
>
> ## References
>
> * Sutton, R. S. and Barto, A. G. (2018). Reinforcement learning: an introduction.
> * Mnih, V., Kavukcuoglu, K., Silver, D., Rusu, A. A., Veness, J., Bellemare, M. G., Graves, A., Riedmiller, M., Fidjeland, A. K., Ostrovski, G., Petersen, S., Beattie, C., Sadik, A., Antonoglou, I., King, H., Kumaran, D., Wierstra, D., Legg, S., and Hassabis, D. (2015). Human-level controlthrough deep reinforcement learning.
> * Naik, A., Wan, Y., Tomar, M., Sutton, R. S. (2024). Reward centering.
> * Lee, H., Lee, Y., Seno, T., Kim, D., Stone, P., Choo, J. (2025). Hyperspherical normalization for scalable deep reinforcement learning.

---

### Author Response · Authors · 2025-12-03
**Summary of Discussion (1/2)**

We thank all of the reviewers for their feedback and extend our appreciation to the Area Chair, acknowledging the extraordinary circumstances that led to an increased workload around this conference. Unfortunately, we did not receive much further response from the reviewers in over a week following our rebuttal. Nevertheless, we believe that we were able to address most of the concerns raised, as they either requested clarification or believed something contradicted by the main text or prior published work. We have uploaded a revision that aims to clarify some of the misunderstandings and includes a convergence proof to further validate our work's correctness. Below, we summarize the reviews we received and our responses to them.

## Strengths

All reviewers generally agree that the paper is well written, and despite inconsistent soundness scores, they agree that what we presented is correct—none of the reviews identified any unsoundness in our work. Further, most reviewers (LPfW, rykv, QdqX) acknowledge that the method appears effective.

## Weaknesses

**Many reviewers questioned why we would want to extend differential TD/reward centering to the episodic setting (LPfW, Sedk, QdqX), with some noting that episodic returns are already bounded. There were also concerns around the extent of the contribution (Sedk, QdqX).**

We emphasize that our primary motivation comes from two papers by Elsayed et al. (2024) and Naik et al. (2024). Elsayed et al. (2024) showed how crucial normalization is for enabling streaming deep reinforcement learning to perform on par with—and sometimes better than—popular non-streaming algorithms (e.g., PPO, SAC). Naik et al. (2024) extended differential TD to discounted objectives—*a setting that already does not require centering for boundedness*—and demonstrated that removing a large state-action-independent offset in the value function improves sample efficiency in infinite-horizon, continuing tasks.

However, Naik et al.’s work cannot be applied to episodic problems because it does not guarantee preserving the optimal policies. The theoretical problem we address is thus how to soundly center rewards while preserving the original ordering of policies, and not how to soundly bound returns. This gap in the literature and whether people consider reward centering is rather explicit: for example, Lee et al. (2025) state that they “do not center the reward, as shifting the reward can alter the optimal policy in episodic tasks.” We believe this highlights a meaningful and relevant gap in the literature and shows that our contribution—while it may appear simple in hindsight, with the simplicity being a plus—is well motivated and well situated within it. We further handle the $\gamma = 1$ extreme, which—emphasized by needing separate cases in the example pseudocode—needs additional care to be sound.

As additional motivation, we note that in the SGD literature, centering is known to reduce gradient covariance and improve sample efficiency if the residuals have a non-zero mean (e.g., due to representational capacity, biased targets as in TD, regularization, etc.). Supervised learning practitioners generally normalize inputs and outputs using statistics computed over their datasets. In reinforcement learning, prior work (e.g., Elsayed et al., 2024) has explored online normalization of inputs and outputs. However, output centering has been omitted because it can change the underlying optimization problem—our work directly addresses this.

**Several reviewers (LPfW, QdqX) notes that we lacked theoretical analysis (e.g., around whether our algorithm converges).**

We acknowledge that the original submission did not contain explicit theorems. However, our central analysis demonstrates: 1) centering can be done while preserving policy ordering, and 2) differential TD (with discounting or terminal states) is exactly equivalent to a particular choice of feature representation. While verbalized informally, this exact equivalence is crucial because it connects differential TD to a class of algorithms with known convergence guarantees. To further validate our conclusions, the revised submission includes an explicit convergence proof in the appendix.

---

> ### Author Response · Authors · 2025-12-03
> **Summary of Discussion (2/2)**
>
> **Reviewer LPfW expressed confusion around the reparameterization presented in Section 4.**
>
> We emphasize that the reparameterization is introduced as a definition, not something derived. It is not meant to suggest how to set parameters, but rather to show that there exists a direct mapping under which an instance of differential TD is equivalent to adding a state-action-independent bias unit.
>
> Naik et al. (2024) noted that differential TD “may seem similar to, but is different from, a value-function unit with a bias weight.” Our reparameterization shows that it is exactly equivalent. We also clarify that this should not be confused with including a bias feature in the final linear layer of a neural network. This is generally not the same as each output (e.g., each action-value) would then have its own bias weight, impacting structural credit assignment.
>
> **Reviewer Sedk believed that 1) we compared against outdated methods, 2) reward centering is complex, and 3) benefits may diminish with other improvements. Reviewer rykv also thought we only considered the value-based setting.**
>
> We extended *streaming* Q($\lambda$) and *streaming* AC($\lambda$) (Elsayed et al., 2024). We acknowledge the confusion, but these are *not* vanilla Q-learning or AC and are *more recent* than any method cited by Reviewer Sedk. We also argue that reward centering is not complex—it introduces only a single scalar and can be easily integrated into any algorithm, as Naik et al. (2024) emphasized. The additional improvements mentioned by Reviewer Sedk have previously been combined with reward centering (Naik et al., 2024), albeit only in infinite-horizon settings, providing evidence that these benefits are orthogonal and not diminished.
>
> **Reviewer QdqX claims that the differential versions of Q-Learning/Stream Q/Stream AC show no distinct advantage over their vanilla counterparts, as the final converged rewards are evidently very similar.**
>
> We disagree with this. Even when final performance appeared similar, the differential variants often reached that performance faster and were never slower when they did not. This is especially important in the streaming setting, where updates occur on every transition in contrast with large, periodically processed batches to produce a frozen, final solution. Because our method adds negligible computation (Reviewer rykv) and streaming algorithms generally use less compute than batch-update methods (e.g., PPO, SAC), improvements in sample efficiency translate directly to compute efficiency (for the same or better final performance).
>
> We further highlight that our evaluation used 30 independent runs, improving statistical confidence and reproducibility (Patterson et al., 2023), in contrast with the typical 3–5 seeds used in the deep RL literature.
>
> ## References
>
> * Naik, A., Wan, Y., Tomar, M., Sutton, R. S. (2024). Reward centering.
> * Elsayed, M., Vasan, G., Mahmood, A. R. (2024). Streaming deep reinforcement learning finally works.
> * Lee, H., Lee, Y., Seno, T., Kim, D., Stone, P., Choo, J. (2025). Hyperspherical normalization for scalable deep reinforcement learning.
> * Patterson, A., Neumann, S., White, M., White, A. (2023). Empirical design in reinforcement learning.

---

### Meta-Review · Area_Chair_qemz · 2026-01-12

**Summary:**

The paper studies extension of differential TD learning to episodic case. In the infintie horizon problem, average reward is often estimated and subtracated in the TD learning udpate, which becomes a problem in episodic case as subtracting constant reward can cause early termination or non-terminating behavior.

The concerns of the reviewers include (i) lack of theoretical analysis, (ii) limited significance and unclear motivations, and (iii) lack of novelty (i.e., tihs works mostly builds on existing principles (potential-based reward shaping, bias-augmented TD, and differential TD) rather than introducing fundamentally new algorithms or analyses.

**Reviewer Concerns:**

Although the authors later clarified that their central analysis connects differential TD to a class of algorithms with known convergence guarantees and included an explicit convergence proof in the appendix, I believe concerns on the significance and novelty of the work still remain valid after the rebuttal. Given the high bar of ICLR, I would suggest rejection and advise the authors to better motivate the setting and explain the novelty of their approach in the next revision.

**Reviewer Scores:**

I suspect the two reviewers with a score of 2 would possibly increase their scores to 4. However, it is hard to imagine that all the reviewers increase their scores to positive ones.

---

### Decision · Program_Chairs · 2026-01-26

Reject